# Molecular and Genetic Biomarkers in Idiopathic Pulmonary Fibrosis: Where Are We Now?

**DOI:** 10.3390/biomedicines11102796

**Published:** 2023-10-16

**Authors:** Ioannis Tomos, Ioannis Roussis, Andreas M. Matthaiou, Katerina Dimakou

**Affiliations:** 15th Department of Respiratory Medicine, ‘SOTIRIA’ Chest Diseases Hospital of Athens, 11527 Athens, Greece; giannisroussis5@gmail.com (I.R.); matthaiou.andreas@gmail.com (A.M.M.); kdimakou@yahoo.com (K.D.); 2Laboratory of Molecular and Cellular Pneumonology, Medical School, University of Crete, 714 09 Heraklion, Greece; 3Respiratory Physiology Laboratory, Medical School, University of Cyprus, Nicosia 2029, Cyprus

**Keywords:** idiopathic pulmonary fibrosis, acute exacerbation of idiopathic pulmonary fibrosis, molecular biomarkers, genetics, diagnostic, prognostic role, peripheral blood

## Abstract

Idiopathic pulmonary fibrosis (IPF) represents a chronic progressive fibrotic interstitial lung disease of unknown cause with an ominous prognosis. It remains an unprecedent clinical challenge due to its delayed diagnosis and unpredictable clinical course. The need for accurate diagnostic, prognostic and predisposition biomarkers in everyday clinical practice becomes more necessary than ever to ensure prompt diagnoses and early treatment. The identification of such blood biomarkers may also unravel novel drug targets against IPF development and progression. So far, the role of diverse blood biomarkers, implicated in various pathogenetic pathways, such as in fibrogenesis (S100A4), extracellular matrix remodelling (YKL-40, MMP-7, ICAM-1, LOXL2, periostin), chemotaxis (CCL-18, IL-8), epithelial cell injury (KL-6, SP-A, SP-D), autophagy and unfolded protein response has been investigated in IPF with various results. Moreover, the recent progress in genetics in IPF allows for a better understanding of the underlying disease mechanisms. So far, the causative mutations in pulmonary fibrosis include mutations in telomere-related genes and in surfactant-related genes, markers that could act as predisposition biomarkers in IPF. The aim of this review is to provide a comprehensive overview from the bench to bedside of current knowledge and recent insights on biomarkers in IPF, and to suggest future directions for research. Large-scale studies are still needed to confirm the exact role of these biomarkers.

## 1. Introduction

Idiopathic pulmonary fibrosis (IPF) represents the most common idiopathic interstitial pneumonia. It occurs mainly in older age and constitutes a chronic, irreversibly progressive, fibrotic interstitial lung disease of unknown aetiology. It is characterized by the presence of the radiological and/or histological pattern of usual interstitial pneumonia (UIP). Unfortunately, despite the recent progress in treatment and the introduction of antifibrotics, IPF still represents a lethal disease with poor prognosis. Its clinical course remains unpredictable with some patients presenting rapid decline in lung function, while others progressing much more slowly over time [1,2,3]. Each year, approximately 5% to 10% of IPF patients develop an acute exacerbation, which is characterized by an unexplained worsening of dyspnoea accompanied by severely impaired gas exchange and new radiographic infiltrates that can lead to their death [1]. An acute exacerbation of idiopathic pulmonary fibrosis (AE-IPF) occurs mainly in more severe cases and represents a major cause of morbidity and mortality in these patients [4,5]. It is estimated that approximately 40% of all deaths are due to AE-IPF [6].

The clinical presentation of IPF includes unexplained exertional dyspnoea and chronic dry cough [7]. The physical examination reveals Velcro-like crackles on auscultation, while the typical radiological image on the high-resolution chest computed tomography (HRCT) consists of the presence of the usual interstitial pneumonia (UIP) pattern, characterized by honeycombing (subpleural, well-defined-walled, cystic airspaces). The pathogenesis of IPF is considered to be epithelial-driven [8]. A repetitive alveolar epithelial injury in genetically susceptible individuals is considered responsible for the activation of an aberrant wound-healing process, excessive production of pro-fibrotic mediators, activation of fibroblasts and accumulation of the extracellular matrix (ECM) [8,9,10,11,12].

IPF represents today a major clinical challenge due to its unknown aetiology and its irreversibility. In such a challenging disease, identifying qualified diagnostic, prognostic and predisposition biomarkers, and reliable indicators of pathogenic processes, would represent a crucial tool in the management of IPF. These markers could reduce uncertainty in early diagnosis and provide accurate predictions and disease efficacy, thus changing the game in the disease course. Even though several biomarker tests are under investigation in IPF, so far none are in widespread clinical practice. In addition, lately, genetics have emerged as a promising clinical tool in IPF [13]. Genetic variants, or genotypes, constitute, therefore, biomarkers with potential diagnostic, prognostic and predisposition value for pulmonary fibrosis. Impaired mucus, surfactant homeostasis and telomere maintenance are some of the hallmarks of genetics regarding IPF unravelling the interaction between environmental injuries, predisposition and related extrinsic and intrinsic offending factors [13]. Predisposition biomarkers could therefore constitute a critical tool for informative counselling, prompt preventive measures and early pulmonary fibrosis detection and treatment [14]. In this review, we will try to elucidate novel insights in the era of molecular and genetic biomarkers in IPF. The main molecular biomarkers presented in this review are summarised in Table 1.

### Molecular Biomarkers

Biomarkers are generally defined as a measurement of any molecule that reflects a disease process or response to a therapeutic intervention [15]. Diverse categories have been developed so far, such as diagnostic (used to detect a disease), prognostic (to identify progression during disease), response biomarkers (to reveal any biological response after the initiation of treatment) and susceptibility/risk biomarkers (to indicate the potential risk for developing a disease in individuals with no clinically apparent disease). The potential sources of biomarkers include peripheral blood, bronchoalveolar lavage fluid (BALF) and lung tissue [14,15].

**Table 1 biomedicines-11-02796-t001:** Diagnostic and prognostic molecular biomarkers in IPF. The symbol ‘+++’ is used when there are lots of evidence and promising data, supportive for the potential role of the biomarker in the literature, while ‘++’ is used when less promising data exists and ‘+’ when there is contradicted data. The symbol ‘-’ is used when there is no supportive evidence for such a potential role so far. Abbreviations: S100A4: S100 calcium-binding protein A4; cCK-18: caspase-cleaved cytokeratin-18; KL-6: Krebs von den Lungen-6; YKL-40: chitinase 3-like protein 1; MMP-7: matrix metalloproteinase-7; ICAM1: intercellular adhesion molecule 1; SP-A: surfactant protein A; SP-D: surfactant protein D; LOXL2: lysyl oxidase-like 2; CCL18: CC chemokine ligand 18; IL-8: interleukin 8; OPN: osteopontin; AECs: alveolar epithelial cells.

Biomarker	Pathogenetic Process	Diagnostic	Prognostic	Specimen	References
S100A4	Fibrogenesis	++	++	Serum BALF Lung tissue	[16,17,18,19,20,21]
cCK-18	AECs apoptosis	+	-	Serum	[22]
KL-6	Alveolar epithelial marker	+	+++	Blood BALF	[23,24,25,26,27,28]
YKL-40	Adhesion molecule	-	++	Blood BALF Lung tissue	[29,30]
MMP-7	Extracellular remodelling	++	+++	Blood BALF Lung tissue	[31,32,33]
ICAM1	Adhesion molecule	-	++	Serum Blood	[34,35]
SPA & SPD	Alveolar epithelial markers	-	++	Serum	[36,37,38,39]
LOXL2	Extracellular matrix remodelling Fibrogenesis	-	+	Serum Lung tissue	[40,41]
Periostin	Extracellular matrix remodelling Fibrogenesis	-	+	Serum Lung tissue	[42,43]
CCL-18	Alternative alveolar macrophage activation	-	+	Serum BALF	[44,45,46]
IL-8	Potent chemotactic activity for polymorphonuclear leukocytes	-	+	Blood BALF	[47,48,49]
OPN	Inflammation Fibroblast migration and proliferation	-	+	Serum BALF Lung tissue	[50,51]
LC3β	Autophagy	-	+	Lung Tissue	[52,53]
BiP, XBP1	Unfolded protein response	-	+	Lung Tissue	[52,54]

## 2. Diagnostic Biomarkers

Several molecular diagnostic biomarkers have been investigated so far in diverse biological specimens, including peripheral blood, serum or plasma and BALF (Table 1).

### 2.1. S100 Calcium-Binding Protein A4 (S100A4)

S100 calcium-binding protein A4 (S100A4) belongs to the S100 superfamily of intracellular binding proteins regulating diverse cellular processes during fibrogenesis [16,17]. It represents a marker of fibroblasts through diverse actions [18]. Scientific evidence suggests that there is a role not only in promoting the transition of fibroblasts to myofibroblasts through the induction of α-smooth muscle actin and collagen 1, but also in contributing to mesenchymal cell shapes though cytoskeletal membrane interactions with myosin II, tropomyosin, actin, and tubulin [17,18,55,56,57]. S100A4 has been widely used as a marker of lung fibroblast identification in experimental studies, such as in bleomycin-induced lung fibrosis animal models [19]. Regarding IPF, S100A4 mRNA and protein levels have been found to significantly increase in BALF compared to controls and other interstitial lung diseases (ILDs) [20,21], but also in lung tissue, as shown by immunohistochemistry studies [16,18]. In the above studies, abundant S100A4-expressing cells are generally described in areas adjacent to mature fibrosis and fibroblastic foci reflecting lung fibrosis activity [18]. Last, but not least, higher serum levels are associated with rapid progression and higher mortality in IPF patients supporting a promising prognostic role of the biomarker [18]. S100A4 seems to offer good diagnostic accuracy with high specificity and sensitivity for the differential diagnosis of IPF among other ILD patients [20]. All these data suggest that S100A4 may represent a promising diagnostic and prognostic biomarker in IPF which deserves to be further investigated.

### 2.2. Caspase-Cleaved Cytokeratin-18 (cCK-18)

Caspase-cleaved cytokeratin-18 (cCK-18) is a circulating cleaved fragment of cytokeratin-18 (CK-18), a cytoskeletal protein found in alveolar epithelial cells (AECs). It has been proposed that cCK-18 is produced during AEC apoptosis [14]. A recent study investigated its diagnostic and prognostic role and suggested that IPF patients present significantly increased serum cCK18 levels compared to healthy controls, while its prognostic role in IPF was not confirmed [22]. However, further studies are necessary due to the currently limited data.

## 3. Prognostic Biomarkers

### 3.1. Krebs von den Lungen-6 (KL-6)

Krebs von den Lungen-6 (KL-6)/mucin 1 (MUC1) represents one of the most extensively studied biomarker in IPF. It constitutes a glycoprotein, expressed on the extracellular surface of type 2 alveolar epithelial cells (AEC2s) and of bronchiolar epithelial cells reflecting alveolar epithelial damage [23,24]. Indeed, it has been previously shown that an increased expression is found in injured and proliferating AEC2s, a mechanism that also underlies in IPF, as the disease is considered epithelial-driven, whereby an abnormal wound-healing process follows [8,25]. In particular, in IPF, KL-6 is detected not only in peripheral blood but also in BALF [26]. Its high blood concentrations have been associated with decreased IPF survival in multiple studies [14,23,24,58], while it has been proposed to represent an independent predictive factor of an acute exacerbation of IPF (AE-IPF). KL-6 serum levels higher than 1000 U/mL have been proposed to be associated with worse prognoses, while levels more than 1300 U/mL indicate an increased risk for developing AE-IPF [27,28,59]. Furthermore, in their recent study, Bennett D and colleagues showed that KL-6 was correlated with markers of disease severity in IPF, such as the diffusing capacity of the lungs for carbon monoxide (DLCO) and oxygen treatment [26]. A diagnostic role has also been proposed. However, most data are based on small-sized studies which lack replicability [58]; the fact that the data reflect alveolar epithelial damage makes them less specific for IPF [14,58]. Indeed, there is emerging evidence that KL-6 is also increased in other f-ILDs, such as chronic hypersensitivity pneumonitis, collagen tissue disorder-related ILD and pulmonary alveolar proteinosis [60,61,62]. Therefore, KL-6 seems to be a promising, useful biomarker in the severity stratification and prognosis of IPF, but not so useful in its diagnosis.

### 3.2. Chitinase 3-like Protein 1 (YKL-40)

The chitinase 3-like protein 1 (YKL-40) is another biomarker whose increased levels in serum and BALF are associated with poor survival in IPF. This molecule has been implicated in diseases, in which fibrosis, inflammation and tissue remodelling prevail [29]. In IPF, increased YKL-40 expression is detected in lung tissues adjacent to fibrotic lesions, in alveolar macrophages, and in bronchiolar epithelial cells [30]. Serum levels higher than 79 ng/mL have been proposed to be associated with worse prognoses [29]. As shown in their retrospective study, Korthagen and co-authors, by using a cut-off ratio of serum YKL-40, managed to distinguish two categories of IPF patients with district survival characteristics, further enforcing its potential prognostic role [29]. Finally, its diagnostic utility is in doubt due to a lack of diagnostic specificity, lack of replication and small study size [62].

### 3.3. Matrix Metalloproteinase-7 (MMP-7)

Matrix metalloproteinase-7 (MMP-7) represents a pluripotent matrix metalloprotease involved in the remodelling of the extracellular matrix, expressed by AEC2s. In general, MMPs have been proposed to act as positive regulators of fibrogenesis in several studies [10,31,32]. MMP-7 has been proposed to reflect the activity of the WNT/β-catenin pathway, a well-known implicated pathway in the pathogenesis of IPF [33,63,64]. Increased MMP-7 levels are detected in lung tissue, BALF and peripheral blood in IPF patients [33]. Together with MMP-1, another metalloprotease, they were included in a combinatorial peripheral blood signature as crucial components [33]. In their study, Rosas IO and colleagues revealed that when combined with these two MMPs, their concentrations can discriminate IPF patients from healthy controls and other ILDs including hypersensitivity pneumonitis, a disease that may present with similar clinical and radiological images to IPF. Moreover, the increased plasma levels of both biomarkers are shown to be correlated with disease severity, as indicated by forced vital capacity (FVC) and DLCO [33]. Besides the diagnostic role of MMP-7, its high blood concentration is also proposed to predict mortality, as shown in a large follow-up study of 140 IPF patients and confirmed in a subsequent independent validation cohort with 101 patients [63]. In particular, levels higher than 4.3 ng/mL have been proposed to be related with IPF progression and worse survival [33]. In addition, as emerged from the Bosentan Use in Interstitial Lung Disease (BUILD)-3 trial, baseline levels of MMP-7 were the only reliable predictor of lung function decline and IPF progression [64]. Last but not least, the aforementioned study assessed, for the first time, longitudinal MMP-7 level changes over time, revealing that such changes may follow disease worsening [64]. Moreover, other MMPs, such as MMP-2, MMP-9, MMP-14, and MMP-19, have also been investigated as diagnostic and prognostic biomarkers in IPF [10,31,32,65]. Indeed, MMP-2, but also MMP-14, were found to be strongly up-regulated in IPF lung tissue compared to non-IPF patients [65]. Increased levels of active MMP-2 have been also reported in BALF from IPF patients [65]. Similarly, MMP-9 has also been found to be highly up-regulated in IPF lungs, specifically in myofibroblast foci, enhancing fibroblast migration [66,67].

### 3.4. Tissue Inhibitors of Metalloproteinases (TIMPs)

Tissue inhibitors of metalloproteinases control the catalytic action of MMPs and consist of four members (TIMP1-TIMP4) [68,69]. Their exact role in the pathogenesis of IPF, but also as potential diagnostic or prognostic biomarkers, still remains unclear [69]. TIMP levels have been found to be up-regulated in IPF lungs compared to controls. Particularly, TIMP-2 seems to be overexpressed in myofibroblast foci, while TIMP-3 is found in the thickened alveolar space [69]. It has been proposed that the TGF-b pathway interferes with TIMP-1 signalling [66,69]. However, further studies are necessary to unravel the exact role of TIMPs in lung fibrosis.

### 3.5. Intercellular Adhesion Molecule 1 (ICAM-1)

Intercellular adhesion molecule 1 (ICAM-1) represents a cell surface glycoprotein that has a key role in cellular function, adhesion-dependent signalling and tissue integrity [70]. As an adhesion molecule, ICAM-1 participates in leukocyte rolling and the crossing of the endothelial layer during inflammation [71]. Besides mediating cell-to-cell interactions, ICAM-1 is also responsible for triggering intracellular signalling events through interactions with various signalling proteins and growth factor receptors, thus modulating cellular responses in response to the surrounding microenvironment [70,71]. In IPF, ICAM-1 may serve as a potential marker of endothelial injury [63]. Increased ICAM-1 levels have been observed in the peripheral blood of IPF patients, while values higher than 202.5 ng/mL seem to be associated with worse prognoses [34,35,59,63]. Recently, in a large study which used data from three independent IPF cohorts, the prognostic role of ICAM-1 serum levels emerged among other proteins, including MMP-7, OPN and periostin, in identifying IPF patients at risk of progression at 12 months [50].

### 3.6. Surfactant Proteins SP-A and SP-D

Surfactant proteins constitute lipoprotein complexes that are produced and secreted by AEC2s. They are composed of 90% lipids and 10% proteins, which is crucial in lung homeostasis. Surfactant-associated protein (SP)-A, a collectin, is constituted as the most abundant among surfactant proteins. It presents hydrophilic properties and participates with SP-D in an innate immune response. Mutations in genes encoding SP-A, *SFTPA1* and *SFTPA2* have been so far associated with familiar pulmonary fibrosis and rarely with sporadic IPF [72,73]. Moreover, increased levels of SP-A and SP-D in serum are detected not only in IPF but also in other fILDs, therefore limiting their specific diagnostic role [36,37,74]. Last, but not least, emerging data support their prognostic value in IPF with both being predictors of worse survival [38,39]. Specifically, an increase of 49 ng/mL in the baseline SP-A levels is associated with an increased risk of mortality in the first year after diagnosis [38,39,59].

### 3.7. Lysyl Oxidase-like 2 (LOXL2)

Lysyl oxidase-like 2 (LOXL2) represents an extracellular matrix crosslinking enzyme known to critically contribute to fibrogenesis [75]. Activated fibroblasts are capable to secrete LOXL2, leading to extracellular matrix remodelling and the release of latent transforming growth factor (TGF)-b [76]. In IPF, increased LOXL2 protein expression is detected in lung tissue, mainly in fibroblastic foci and collagenous regions [40]. Furthermore, its elevated serum levels are associated with an increased risk of progression in IPF patients, especially when they exceed the cut-off of 700 pg/mL [41]. In this large study, despite the fact that LOXL2 serum levels were not correlated with disease severity, as indicated by lung function impairment values, it was proposed that they may reflect IPF disease activity [41]. However, it should be mentioned that the anti-LOX2-targeted therapy failed to reduce disease progression [59].

### 3.8. Periostin

Periostin constitutes an extracellular matrix protein implicated in pulmonary fibrosis. Up-regulated by TGF-b, it contributes to extracellular matrix deposition as well as mesenchymal cell proliferation [42]. IPF fibroblasts are able to produce periostin and its increased levels are indeed observed in lung tissue, mainly in fibroblastic foci [42]. Moreover, increased levels are also detected in peripheral blood with a prognostic value and are associated with disease progression [43]. Recently, its prognostic role was also confirmed by a large IPF cohort assessing biomarker profiles in progressive IPF [50]. Last but not least, periostin has also been investigated in experimental studies, in bleomycin-induced animal models, in which periostin-deficient (periostin^−/−^) mice showed protection from fibrosis development [42].

### 3.9. CC Chemokine Ligand 18 (CCL18) 

The main source of CC chemokine ligand 18 (CCL18) in the lungs is alveolar macrophages. CC chemokine ligand 18 represents an indicator of the alternative macrophage activation. Indeed, Prasse and co-authors have already shown that CCL18 gene expression and protein production are up-regulated in normal alveolar macrophages (AMs) after Th2 cytokine stimulation [44]. They also revealed positive feedback between AMs and fibroblasts in promoting collagen production, as culture supernatants of AMs from IPF patients led to increased collagen production by normal lung fibroblasts through CCL18 release. Indeed, expressed CCL18 levels were significantly increased in BAL-derived cells from IPF patients compared to patients with other fibrotic diseases, such as sarcoidosis or hypersensitivity pneumonitis, and healthy controls [44]. The aforementioned data suggest that CCL18 may have a diagnostic role in discriminating IPF from healthy controls. However, its diagnostic utility in identifying IPF is limited, as Cai M and co-authors revealed that CCL18 serum levels are also significantly elevated in other ILDs, such as hypersensitivity pneumonitis [77].

Moreover, serum CCL18 levels have been proposed to exhibit a prognostic role in IPF patients, as increased levels correlate with the risk of AE-IPF, poor survival and lung function decline [45]. Recently, it has been proposed that specific genetic variations in the CCL18 gene lead to increased CCL18 mRNA and protein expressions, thus predisposing them to poor prognoses in IPF [46].

### 3.10. Interleukin 8 (IL-8)

Interleukin 8 (IL-8) is a chemo-attractant cytokine for neutrophils [78]. It is released by activated mononuclear macrophages and has a potent chemotactic activity for polymorphonuclear leukocytes [47]. IL-8 is found elevated in the BALF and serum of IPF patients, reflecting disease activity. Its levels are also correlated negatively with pulmonary function tests and survival [47], while increased levels during AE-IPF are significantly associated with worse outcomes [48]. The increased detected levels in BALF indicate an underlying progressive phase during the course of the disease [49].

### 3.11. Osteopontin

Osteopontin (OPN) represents another biomarker whose diagnostic and prognostic role has been investigated in IPF. It is a phosphorylated glycoprotein, a mediator of inflammation, immune response and wound healing [79]. It is expressed mainly by AECs and activated macrophages, exhibiting profibrotic effects which affect fibroblast migration and proliferation [80]. Increased osteopontin levels have been found in the lungs, BALF and serum of IPF patients, while its serum levels have been proposed to predict mortality [50,51]. However, osteopontin lacks specificity for IPF; it has been found that osteopontin levels have increased in other ILDs but also in obstructive lung diseases, such as chronic obstructive pulmonary disease [81,82]. Therefore, its diagnostic role remains limited in identifying IPF. Last but not least, its potential role in AE-IPF remains unclear with limited available data supporting it so far [83].

### 3.12. LC3β

Autophagy, a lysosome-mediated catabolic process, responsible for degrading damaged organelles and aggregations of proteins, represents an affected survival mechanism against cellular stress in IPF. Decreased autophagic activity is found in IPF lung tissue as being implicated in the epithelial-to-mesenchymal transition and favouring myofibroblasts’ phenotype conversion and production of extracellular matrix components [52,53]. However, the exact mechanism remains unclear and not definitively identified. It seems that TGF-β, the most studied profibrotic mediator affecting IPF fibroblasts and epithelial cells, may inhibit autophagy, increasing a smooth muscle actin and fibronectin expression in fibroblasts [53]. In particular, it has been found that the LC3β expression, an autophagosome marker, has decreased in IPF lung tissue compared to the controls [52]. Moreover, a recent observation study showed an increased accumulation of autophagosomes (LC3β puncta) in IPF lung tissue compared to controls, confirming previous data regarding the dysregulated autophagy in IPF [52]. This autophagy marker was co-localised with unfolded protein response (UPR) markers and associated with improved lung function parameters in IPF patients [52]. However, further studies are necessary to elucidate the exact cross-talk between the diverse cell stress responses implicated in IPF and investigate the presence or not of autophagy biomarkers in the peripheral blood.

### 3.13. BiP and XBP1

The potential association between endoplasmic reticular (ER) stress and IPF development has been extensively studied in recent years [84]. It is hypothesized that epithelial ER stress in IPF represents the consequence of dysregulated autophagy and the subsequent misfolding of aberrant proteins [84,85]. Therefore, the increased accumulation of protein aggregates worsens ER stress, resulting in further epithelial cell death. Indeed, it has been shown that markers of ER stress, such as BiP and XBP1, are elevated in the lung tissue of IPF patients [52,54]. In particular, BiP, an ER chaperon that participates in the docking of UPR sensors, and spliced XPB1, a transcription factor responsible for ER-associated degradation, are considered reliable activation markers of UPR [54]. So far, diverse studies have shown increased expression of both molecules in IPF lung tissue [54,86], while a recent study revealed a correlation with worse lung function, linking for the first time UPR markers with progression in IPF [52]. However, so far, the exact and complex mechanisms by which ER stress promotes fibrosis remain unclear and need further clarification.

## 4. Predisposition Biomarkers

Genetic studies of IPF have improved so far our understanding of the pathways involved in this devastating disease [13]. Genetics represent a promising clinical tool in IPF [13]. Genetic variants, or genotypes, constitute, therefore, potential biomarkers with diagnostic, prognostic and predisposition value for pulmonary fibrosis [87]. Predisposition biomarkers represent a useful, promising tool for informative counselling, prompt preventive measures and early pulmonary fibrosis detection [14]. Several germline mutations have been already associated with pulmonary fibrosis, linked with diverse underlying pathogenetic processes, such as impaired mucus, surfactant homeostasis and telomere maintenance [13]. So far, the causative mutations in pulmonary fibrosis cases include mutations in telomere-related genes (TRGs) and surfactant-related genes (SRG) [87,88]. These genetic markers may also have a role as predisposition biomarkers in IPF.

### 4.1. Telomere Maintainance

Telomeres constitute DNA–protein structures that are located at the chromosome ends and preserve genetic information [11,87,89]. In each cell division, telomeres shorten, ultimately leading to apoptosis and cell cycle arrest when reaching a critical point [87,89]. This telomere shortening is therefore linked with aging [87]. Specialized polymerase, a telomerase that protects TL by adding telomere repeats—six nucleotides (TTAGGG)—to the ends of chromosomes, plays a critical role in restoring telomere loss [89]. Telomerase consists of two essential components: the telomerase reverse transcriptase (hTERT), a catalytic component, and an RNA component (hTR), the template for nucleotide additions by hTERT [90]. In patients with familial pulmonary fibrosis (FPF), mutations in either of the essential components of telomerase have been found to contribute to an acceleration of telomere shortening [91].

Telomere-related gene (TRG) mutations were the first to be identified in FPF. In particular, mutations of telomere-related genes represent the most frequent in pulmonary fibrosis, with approximately 15% of familial cases [92]. Usually, these patients with pulmonary fibrosis, carriers of TRG mutations, present with a UIP pattern on HRCT; however, other forms of fibrotic ILDs are also possible, including a non-specific interstitial pneumonia pattern, or unclassifiable fibrosis and a combined pulmonary fibrosis and emphysema (CPFE) pattern [87].

Telomere length has been proposed as a potential biomarker in IPF as it is measured in peripheral blood. It is generally expressed as a relative length compared to a control population and corrected for age [87]. The two widely accepted and available techniques include real-time quantitative PCR (qPCR) and flow cytometry (flow-FISH) [93]. The latter, although more expensive and technically demanding, provides more reproducible results [93]. Short telomeres express the limited tissue renewal capacity in the lung and represent an important risk factor for the development of IPF [11,94,95]. Recent studies have shown that carriers of TRG mutations have a shorter telomere length compared to age-matched controls [96]. Last but not least, short telomeres are also associated with poor survival in IPF [97]. Therefore, the telomere length represents a potential biomarker, not only to screen patients with TRG mutations, but also in general in IPF [11,87,97].

### 4.2. Surfactant-Associated Genes

Mutations of surfactant-associated genes, including *SFTPA1* (location: 10q22.3)*, SFTPA2* (10q22.3)*, SFTPC* (8p21.3)*, SFTPB* (2p11.2)*, ABCA3* (16p13.3) and *NKX2.1* (14q13.3)), are the second most frequent forms of pulmonary fibrosis, approximately occurring in 1–5% of familial cases [92]. Surfactant is produced and secreted by AEC2s. It is composed of 90% lipids and 10% proteins, which is crucial in lung homeostasis [98]. They are encoded by *SFTPA, SFTPB, SFTPC*, and *SFTPD* genes. Surfactant-associated protein (SP)-A, a collectin, is the most abundant among surfactant proteins, presenting hydrophilic properties and participating with SP-D in an innate immune response. On the other hand, the hydrophobic SP-B and SP-C contribute to surfactant function. Lately, mutations in the *SFTPA1* and *SFTPA2* have been associated with familiar pulmonary fibrosis and rarely with sporadic IPF [72,73,99]. Lung cancer develops in up to one third of patients carrying *SFTPA1* and *SFTPA2* germline mutations, suggesting an increased risk of lung cancer [72,99]. In addition, *SFTPB* mutations have been so far associated with neonatal respiratory distress syndrome but not with adult pulmonary fibrosis [100], while *SFTPD* mutations have not been associated with lung disease [87]. *SFTPC* mutations have been also associated with the occurrence of ILD in adults, but the frequency remains very rare, with less than 5% of familial cases [87]. Its transmission is autosomal dominant.

Moreover, the ATP-binding cassette family A, member 3 (ABCA3) transporter encoded by *ABCA3*, is responsible for transporting surfactant lipids into lamellar bodies [101]. Recently, bi-allelic missense *ABCA3* mutations have been identified in adult patients with childhood-onset ILD [102,103]. *ABCA3*-mutation-related lung disease inheritance is autosomal recessive, requiring two disease-causing mutations (bi-allelic), one from each parent [102,103].

*NKX2.1* heterozygous mutations have been also associated with ILD development in the context of a triad, known as “brain–lung–thyroid syndrome” [104]. Patients, carriers of mutations in *NKX2.1*, present central nervous system manifestations, such as hypothyroidism ILD [87]. The *NKX2.1* gene codes for thyroid transcription factor 1 (TTF1), a critical molecule for lung development which regulates surfactant protein expressions. The exact pathophysiology remains unknown; however, the prevailed hypothesis is related to endoplasmic reticulum stress and caspase pathway activation in AEC2s.

### 4.3. MUC5B

The common gain-of-function variant rs35705950 in the promoter of MUC5B, which encodes mucin 5B, represents the strongest genetic risk factor, accounting for 30% of the risk of developing IPF [87,105]. MUC5B constitutes a large salivary glycoprotein, a major component of the respiratory mucus. It seems that IPF patients carrying this promoter variant present slower progression and better survival [106]. In these patients, the increased expression of MUC5B, a result of the diminished mucociliary clearance, could either cause further injury or impair the existing vulnerable repair mechanisms in the distal IPF lung [107]. It should be noted though that the presence of this polymorphism has been so far associated with other fibrotic ILDs, such as rheumatoid arthritis-associated ILD, hypersensitivity pneumonitis and asbestosis [108,109,110].

### 4.4. Future Prospects for IPF

The development of validated biomarkers in IPF seems more critical than ever for a future personalized therapy for fibrotic patients. Recognising easy-to-measure peripheral blood biomarkers would facilitate prompt diagnoses, improved prognoses, and therapeutics which foster the rebirth of the natural history of IPF. By incorporating novel emerging genomic techniques and molecular tools in clinical practice, our therapeutic decision-making and treatment efficacy will be enhanced. Therefore, elucidating the role of promising molecules implicated in the underlying complex pathways in IPF, such as fibrogenesis, extracellular matrix remodelling, chemotaxis, epithelial cell injury, autophagy, and unfolded protein response, represents a critical and promising research field that needs to be further investigated. With such advancements, novel drug targets could be developed in the future, this improving our therapeutic options for IPF.

## 5. Conclusions

The emergence of reliable and validated molecular and genetic biomarkers in clinical practice in IPF is becoming increasingly more urgent. Diagnostic, prognostic and predisposition peripheral blood biomarkers could play a critical role in early diagnosis, the accurate stratification of disease severity, and better management during its course. Several molecules have been investigated so far in smaller cohorts, while the recent uthe respiratory mucusse of biomarker index composed of multiple biomarkers has shown a promising potential. Further larger validation studies are necessary to confirm the diagnostic and prognostic utility of these various molecular and genetic biomarkers in the clinical practice of IPF.

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
