# Peer review of "Molecular and Genetic Biomarkers in Idiopathic Pulmonary Fibrosis: Where Are We Now?"

_biomedicines, 2023, doi:10.3390/biomedicines11102796_

Round 1
Reviewer 1 Report
This review by Tomos et al aims to summarize recent advances in identifying biomarkers for idiopathic pulmonary fibrosis. The following is a synopsis of concerns raised based on the data presented in the manuscript.
Major:
1) It is important to include Muc5b polymorphism in the discussion of the genetics of IPF.
2) Some of the references were not studied in IPF populations.
5) It is unclear what “+” , “++”, “+++”, and “-“ in Table 1 mean.
6) It is inappropriate to cite review articles as references in table 1.
7) Extensive English editing is required.
Minor:
1) Please proofread all the references. Most of them were numbered incorrectly.
Extensive editing is required.
Author Response
Reply to comments
biomedicines-2576744
Molecular and genetic biomarkers in Idiopathic Pulmonary Fibrosis. Where are we now?
We thank the Editor and the Reviewers for the careful review of our manuscript and their insightful comments. Following their suggestions, we have performed a substantial revision with additional analyses, which we believe have improved our manuscript. Below, we provide a point-by-point reply to the Editor’s and Reviewer’s comments. Please note that pages’ numbers correspond to the revised version of the manuscript.
Reviewer reports
Reviewer #1:
This review by Tomos et al aims to summarize recent advances in identifying biomarkers for idiopathic pulmonary fibrosis. The following is a synopsis of concerns raised based on the data presented in the manuscript.
Response: We thank the Reviewer for his/her comments.
Major:
Comment 1: “It is important to include Muc5b polymorphism in the discussion of the genetics of IPF.”
Response 1: We appreciate Reviewer’s comment. We have added a paragraph in the section of predisposing biomarkers regarding MUC5B.
Precisely, in page 12, Lines 386-398, a paragraph was added, as follows:
“MUC5B
The common gain-of-function variant rs3570595016 in the promoter of MUC5B, encoding mucin 5B, represents the strongest genetic risk factor accounting for 30% of the risk of developing IPF.75,93 MUC5B constitutes a large salivary glycoprotein, a major component of the respiratory mucus. It seems that IPF patients carrying this promoter variant present slower progression and better survival.94 In these patients, the increased expression of MUC5B, a result of the diminished mucociliary clearance could either cause further injury or impair the existing vulnerable repair mechanisms in the distal IPF lung.95 It should be noted though that the presence of this polymorphism has been so far associated with other fibrotic ILDs, such as rheumatoid arthritis-associated ILD, hypersensitivity pneumonitis and asbestosis.96–98”
Comment 2: Some of the references were not studied in IPF populations.
Response 2: We thank the Reviewer for his comment. Indeed, all references are not exclusively in IPF. We have used a small proportion of studies from other diseases to present globally the characteristics of the biomarkers and the potential underlying mechanisms in which they participate. As an example, we have used the study of Fei F and co-authors, regarding the role of metastasis-induced protein S100A4 in human non-tumor pathophysiologies, which explains very nice the pathophysiology of S100A4 and some of the characteristics that are potentially also found in pulmonary fibrosis.
Comment 3: It is unclear what “+”, “++”, “+++”, and “-“in Table 1 mean.
Response 3: We thank the Reviewer for his comment. To make clearer the use of the symbols in the Table 1, we have added some sentences in the legend to explain it. Precisely, we have added the text as follows ‘The symbol ‘+++’ is used when there is much evidence and promising data supportive for the potential role of the biomarker in the literature, while ‘+’ when less promising or contradicted data exists. The symbol ‘-’ is used when there is no supportive evidence for such a potential role so far.’
Moreover, to be clearer we added abbreviations under the Table 1 (Page 5, lines: 99-104), as follows:
‘Abbreviations: S100A4: S100 calcium-binding protein A4; cCK-18: caspase-cleaved cytokeratin-18; KL-6: Krebs von den Lungen-6; YKL-40: chitinase 3-like protein 1; MMP-7: matrix metalloproteinase-7; ICAM1: intercellular adhesion molecule 1; SP-A: surfactant protein A; SP-D: surfactant protein D; LOXL2: lysyl oxidase-like 2; CCL18: CC chemokine ligand 18; IL-8: interleukin 8; OPN: osteopontin; AECs: alveolar epithelial cells.’
Comment 4: It is inappropriate to cite review articles as references in table 1.
Response 4: We appreciate Reviewer’s comment. We have revised the references in Table 1 and corrected accordingly citing original research articles.
Comment 5: Extensive English editing is required.
Response 5. We appreciate Reviewer comment. We have revised and edited English language in our manuscript.
Minor:
Comment 1: Please proofread all the references. Most of them were numbered incorrectly.
Response 1: Thank you for Reviewer’s comment. We have carefully proofread all references. We have substantially corrected references in Table 1. Moreover, we have erased reference 44 and revised citations.
Reviewer #2:
General Comments: The current review by Ioannis Tomas and colleagues discusses putative diagnostic, prognostic and predisposition biomarkers for the pathogenesis of IPF. The manuscript is well written, interesting and addresses important clinical problems, likely valuable in clinical practice particularly, in diagnosis as well as in evaluation of new treatments for efficacy. It would be interesting, if the authors could include a table listing how many of these biomarkers are currently being used in clinical practice.
Response: We thank the Reviewer for his/her positive comments. Unfortunately, so far there is no use of any biomarker in clinical practice despite the promising data. We have stated that in the introduction, and we believe that this creates a critical need for further studies to elucidate and validate their role in this devastating disease. Further research is necessary to confirm the potential diagnostic, prognostic and predisposition role and enforce evidence aiming to have prompt diagnosis and disease identification in the near future. Precisely, we have written in Introduction, in page 2 in the 3rd paragraph.
‘In such a challenging disease, identifying qualified diagnostic, prognostic and predisposition biomarkers, reliable indicators of pathogenic processes would represent a crucial tool in the management of IPF. These markers could reduce uncertainty in early diagnosis and provide accurate prediction and disease efficacy changing the game in the disease course. Even though several biomarker tests are being under investigation in IPF, few biomarkers are in widespread clinical practice. In addition, lately, genetics have emerged to a promising clinical tool in IPF.13’
Also, we have stated this need in the conclusion:
‘The emergence of reliable and validated molecular and genetic biomarkers in clinical practice in IPF is becoming more and more urgent. Diagnostic, prognostic and predisposition peripheral blood biomarkers could play a critical role in early diagnosis, accurate stratification of disease severity and better management during its course. Several molecules have been investigated so far in smaller cohorts, while the recent uthe respiratory mucusse of biomarker index composed of multiple biomarkers has shown a promising potential. Further larger validation studies are necessary to confirm the diagnostic and prognostic utility of these various molecular and genetic biomarkers in the clinical practice in IPF.’
Reviewer #3:
General Comments: The topic of the article is very hot, because, even with antifibrotic treatment, the prognosis of IPF is still poor. The subject is extensively addressed, but a few ideas about nonprotein biomarkers could be added (mitochondrial biomarkers, microRNA, microbiome analyses maybe). Some studies described also the role of the mucin gene MUC5B in the pathogenesis of pulmonary fibrosis. The article could be publish with minor revisions.
Response: We appreciate Reviewer’s comments. We have added a paragraph in the predisposition biomarkers regarding MUC5B and its role in the IPF pathogenesis. We agree with the Reviewer that it is very important to be included in the manuscript. Precisely, we have added the following text in page 12, Lines 386-398, as follows:
“MUC5B
The common gain-of-function variant rs3570595016 in the promoter of MUC5B, encoding mucin 5B, represents the strongest genetic risk factor accounting for 30% of the risk of developing IPF.75,93 MUC5B constitutes a large salivary glycoprotein, a major component of the respiratory mucus. It seems that IPF patients carrying this promoter variant present slower progression and better survival.94 In these patients, the increased expression of MUC5B, a result of the diminished mucociliary clearance could either cause further injury or impair the existing vulnerable repair mechanisms in the distal IPF lung.95 It should be noted though that the presence of this polymorphism has been so far associated with other fibrotic ILDs, such as rheumatoid arthritis-associated ILD, hypersensitivity pneumonitis and asbestosis.96–98”
Regarding nonprotein biomarkers in IPF, such as mitochondrial biomarkers, microRNA and microbiome analyses, we added two paragraphs with autophagy and ER stress biomarkers that have been detected in lung tissue. We believe that microRNA, long RNA and microbiome analyses represent aspects beyond the scope of this review and necessitate a review focusing on this field. They are indeed very interesting mechanisms; however, they need further clarification. Precisely, we have added in Pages 10-11, lines:302-340, the following text
‘LC3β
Autophagy, a lysosome-mediated catabolic process, responsible for degrading damaged organelles and aggregations of proteins represents an affected survival mechanism against cellular stress in IPF. Decreased autophagic activity is found in IPF lung tissue being potentially implicated in epithelial-to-mesenchymal transition favouring myofibroblasts’ phenotype conversion and production of extracellular matrix components.74,75 However, the exact mechanism remains unclear and not definitively identified. It seems that TGF-β, the most studied profibrotic mediator affecting in IPF fibroblasts as well as epithelial cells, may inhibit increasing a-smooth muscle actin and fibronectin expression in fibroblasts.75 Particularly LC3β expression, an autophagosome marker has been found decreased in IPF lung tissue compared to controls.74 Moreover, a recent observation study showed an increased accumulation of autophagosomes (LC3β puncta) in IPF lung tissue compared to controls confirming previous data regarding the dysregulated autophagy in IPF.74 This autophagy marker was co-localised with unfolded protein response (UPR) markers and associated with improved lung function parameters in IPF patients.74 However, further studies are necessary to elucidate the exact cross-talk between the diverse cell stress responses implicated in IPF and to inestigate the potential presence of autophagy biomarkers in the peripheral blood.
BiP and XBP1
The potential association between endoplasmic reticular (ER) stress and IPF development has been extensively studied during last years.76 It is hypothesized that epithelial ER stress in IPF represents the consequence of dysregulated autophagy and the subsequent misfolding of aberrant proteins.76,77 Therefore, the increased accumulation of protein aggregates worsens ER stress resulting in further epithelial cell death. Indeed, it has been shown that markers of ER stress, such as BiP and XBP1 are elevated in the lung tissue of IPF patients.74,78 Particularly, BiP, an ER chaperon that participates in the docking of UPR sensors and spliced XPB1, a transcription factor responsible for ER-associated degradation, are considered reliable activation markers of UPR.78 So far, diverse studies has shown increased expression of both molecules in IPF lung tissue,78,79 while a latter one revealed correlation with worse lung function, linking for the first time UPR markers with progression in IPF.74 However, so far, the exact and complex mechanisms by which ER stress promote fibrosis remain unclear and need further clarification.’
Finally, we have added in Table 1, the new biomarkers we have included.
Reviewer #4:
General comments: Further discussions are needed, with some issues not having been mentioned. For instance, autophagy, ER stress and the possible role of till-like receptors are not treated, nor reasons given for them to have been omitted. Besides, the article would gain from including some graphics about altered/hypothetically altered functions (e.g. see Volkova et al, J Gerontol A Biol Sci Med Sci. 2012, 67A(3):247–253). It is not discussed what direction the authors think future research should aim at, which would also enliven the review. In brief, some element of originality is essential.
Response: We appreciate Reviewer’s comments. We have included two paragraphs regarding biomarkers that are implicated in autophagy and ER stress (Pages 10-11, lines:302-340), as follows. Initially, we aimed not to mention these complex mechanisms, as so far these biomarkers have been detected in lung tissue of IPF patients and this crosstalk we think needs further clarification. However, we added as proposed three markers of the aforementioned mechanisms. Regarding Toll-like receptors, we think as most studies are based on experimental studies, that further research is necessary in IPF patients.
‘LC3β
Autophagy, a lysosome-mediated catabolic process, responsible for degrading damaged organelles and aggregations of proteins represents an affected survival mechanism against cellular stress in IPF. Decreased autophagic activity is found in IPF lung tissue being potentially implicated in epithelial-to-mesenchymal transition favouring myofibroblasts’ phenotype conversion and production of extracellular matrix components.74,75 However, the exact mechanism remains unclear and not definitively identified. It seems that TGF-β, the most studied profibrotic mediator affecting IPF fibroblasts and epithelial cells, may inhibit autophagy increasing a-smooth muscle actin and fibronectin expression in fibroblasts.75 Particularly LC3β expression, an autophagosome marker has been found decreased in IPF lung tissue compared to controls.74 Moreover, a recent observation study showed an increased accumulation of autophagosomes (LC3β puncta) in IPF lung tissue compared to controls confirming previous data regarding the dysregulated autophagy in IPF.74 This autophagy marker was co-localised with unfolded protein response (UPR) markers and associated with improved lung function parameters in IPF patients.74 However, further studies are necessary to elucidate the exact cross-talk between the diverse cell stress responses implicated in IPF and investigate the presence or not of autophagy biomarkers in the peripheral blood.
BiP and XBP1
The potential association between endoplasmic reticular (ER) stress and IPF development has been extensively studied during last years.76 It is hypothesized that epithelial ER stress in IPF represents the consequence of dysregulated autophagy and the subsequent misfolding of aberrant proteins.76,77 Therefore, the increased accumulation of protein aggregates worsens ER stress resulting in further epithelial cell death. Indeed, it has been shown that markers of ER stress, such as BiP and XBP1 are elevated in the lung tissue of IPF patients.74,78 Particularly, BiP, an ER chaperon that participates in the docking of UPR sensors and spliced XPB1, a transcription factor responsible for ER-associated degradation, are considered reliable activation markers of UPR.78 So far, diverse studies has shown increased expression of both molecules in IPF lung tissue,78,79 while a latter one revealed correlation with worse lung function, linking for the first time UPR markers with progression in IPF.74 However, so far, the exact and complex mechanisms by which ER stress promote fibrosis remain unclear and need further clarification.’
In addition, we have added in Table 1, the new biomarkers we have included and the related citations.
Finally, regarding Reviewer’s comment about future direction in research, we have added a new paragraph in Page 14, Lines: 451-464, as follows.
‘Future prospects for IPF
The development of validated biomarkers in IPF seems more critical than ever for a future personalized therapy of fibrotic patients. Recognising easy to measure peripheral blood biomarkers would facilitate prompt diagnosis, improved prognosis and therapeutics fostering the rebirth of IPF natural history. By incorporating novel emerging genomic techniques and molecular tools in clinical practice, our therapeutic decision-making and treatment efficacy will be enhanced. Elucidating therefore the role of promising molecules, implicated in the underlying complex pathways in IPF, such as fibrogenesis, extracellular matrix remodelling, chemotaxis, epithelial cell injury, autophagy and unfolded protein response, represents critical and promising research field that need to be further investigated. With such advancements, novel drug targets could be developed in the future improving our therapeutic options in IPF.’
Reviewer #5:
General Comments: In the manuscript titled "Molecular and Genetic Biomarkers in Idiopathic Pulmonary Fibrosis: Where Are We Now?" by Ioannis Tomosi and colleagues, the authors have provided a comprehensive overview of the current knowledge and recent insights regarding potential molecular and genetic biomarkers in IPF, spanning from bench to bedside. The manuscript categorizes these biomarkers into four distinct sections: diagnostic, prognostic, response biomarkers, and susceptibility biomarkers. These biomarkers are derived from peripheral blood, bronchoalveolar lavage, and lung tissue. Overall, this intriguing manuscript offers a well-structured review of the challenges associated with IPF. By identifying robust diagnostic, prognostic, and predisposition biomarkers, it addresses the critical need for reliable indicators of pathogenic processes in the management of IPF. The identification of such blood biomarkers is imperative for ensuring prompt diagnosis and early treatment, and it holds promise for the development of novel drug targets to combat IPF's development and progression.
Response: We thank the Reviewer for his/her positive comments.
Minor:
Comment 1: Line 36-39 – Cite reference(s) for the following sentence “Each year, approximately 5% to 10% of IPF patients develop an acute exacerbation…..”
Response: We thank the Reviewer for his/her comment. We have added the citation.
Comment 2: Line 89 – Remove the abbreviation of “α-SMA”, mentioned only one time in the entire manuscript.
Response: We thank the Reviewer for his/her comment. We have removed the abbreviation.
Comment 3: Line 94 – Write the full form of “ILDs” and remove the ILDs expansion in line 321
Response: We have corrected the text accordingly writing the full form of ILDs in Line 94 and removing the ILDs expansion in line 321.
Comment 4: Line 126 – Write the full form of “DLCO” and remove the DLCO expansion in line 156.
Response: We thank the Reviewer for his/her comment. We have corrected the text accordingly.
Comment 5: There are a few abbreviation/expansion-related errors in the manuscript, please check it and correct them.
Response: We appreciate Reviewer’s comment. We have revised abbreviations/expansion and corrected them accordingly. Moreover, we added abbreviation list under Table 1 to make clearer the context. Precisely,
‘Abbreviations: S100A4: S100 calcium-binding protein A4; cCK-18: caspase-cleaved cytokeratin-18; KL-6: Krebs von den Lungen-6; YKL-40: chitinase 3-like protein 1; MMP-7: matrix metalloproteinase-7; ICAM1: intercellular adhesion molecule 1; SP-A: surfactant protein A; SP-D: surfactant protein D; LOXL2: lysyl oxidase-like 2; CCL18: CC chemokine ligand 18; IL-8: interleukin 8; OPN: osteopontin; AECs: alveolar epithelial cells.’
Comment 6: What about the authors' opinion regarding using ECM and differentiation of cells as a biomarker for IPF?
Response: We thank the Reviewer for his/her comment. We believe that ECM components could play a critical role as biomarkers in IPF. We have included major ones, such as LOXL2 and MMP7 and we believe that in near future ECM biomarkers could be included in clinical practice of ILDs. Regarding, differentiation of cells in IPF, it is also an important issue in the disease, however, we think that it could be not covered in this review, but in a special one focusing on this field.

Reviewer 2 Report
The current review by Ioannis Tomas and colleagues discusses putative diagnostic, prognostic and predisposition biomarkers for the pathogenesis of IPF. The manuscript is well written, interesting and addresses important clinical problems, likely valuable in clinical practice particularly, in diagnosis as well as in evaluation of new treatments for efficacy. It would be interesting, if the authors could include a table listing how many of these biomarkers are currently being used in clinical practice.
Author Response

(The authors gave the same response as above.)

Reviewer 3 Report
The topic of the article is very hot, because, even with antifibrotic treatment, the prognosis of IPF is still poor. The subject is extensively addressed, but a few ideas about nonprotein biomarkers could be added (mitochondrial biomarkers, microRNA, microbiome analyses maybe). Some studies described also the role of the mucin gene MUC5B in the pathogenesis of pulmonary fibrosis. The article could be publish with minor revisions.
Author Response

(The authors gave the same response as above.)

Reviewer 4 Report
Further discussions are needed, with some issues not having been mentioned. For instance, autophagy, ER stress and the possible role of till-like receptors are not treated, nor reasons given for them to have been omitted. Besides, the article would gain from including some graphics about altered/hypothetically altered functions (e.g. see Volkova et al, J Gerontol A Biol Sci Med Sci. 2012, 67A(3):247–253). It is not discussed what direction the authors think future research should aim at, which would also enliven the review. In brief, some element of originality is essential.
Author Response

(The authors gave the same response as above.)

Reviewer 5 Report
In the manuscript titled "Molecular and Genetic Biomarkers in Idiopathic Pulmonary Fibrosis: Where Are We Now?" by Ioannis Tomosi and colleagues, the authors have provided a comprehensive overview of the current knowledge and recent insights regarding potential molecular and genetic biomarkers in IPF, spanning from bench to bedside. The manuscript categorizes these biomarkers into four distinct sections: diagnostic, prognostic, response biomarkers, and susceptibility biomarkers. These biomarkers are derived from peripheral blood, bronchoalveolar lavage, and lung tissue. Overall, this intriguing manuscript offers a well-structured review of the challenges associated with IPF. By identifying robust diagnostic, prognostic, and predisposition biomarkers, it addresses the critical need for reliable indicators of pathogenic processes in the management of IPF. The identification of such blood biomarkers is imperative for ensuring prompt diagnosis and early treatment, and it holds promise for the development of novel drug targets to combat IPF's development and progression.
Minor Comments:
Line 36-39 – Cite reference(s) for the following sentence “Each year, approximately 5% to 10% of IPF patients develop an acute exacerbation…..”
Line 89 – Remove the abbreviation of “α-SMA”, mentioned only one time in the entire manuscript.
Line 94 – Write the full form of “ILDs” and remove the ILDs expansion in line 321
Line 126 – Write the full form of “DLCO” and remove the DLCO expansion in line 156
There are a few abbreviation/expansion-related errors in the manuscript, please check it and correct them.
What about the authors' opinion regarding using ECM and differentiation of cells as a biomarker for IPF?
The manuscript is written in a simple and understandable language; however, I recommend minor English editing to enhance the organization of the sentence.
Author Response

(The authors gave the same response as above.)

Round 2
Reviewer 1 Report
The authors have largely been responsive to my comment, but I want to point out one significant error before the manuscript being accepted.
On page 14, the most common Muc5B SNP is rs35705950 not rs3570595016. The authors need to correct this.
Author Response
We thank the Editor and the Reviewers for the careful review of our manuscript and their insightful comments. Following their suggestions, we have performed a substantial revision with additional analyses, which we believe have improved our manuscript. Below, we provide a point-by-point reply to the Editor’s and Reviewer’s comments. Please note that pages’ numbers correspond to the revised version of the manuscript.
Reviewer reports
Reviewer #1:
The authors have largely been responsive to my comment, but I want to point out one significant error before the manuscript being accepted.
On page 14, the most common Muc5B SNP is rs35705950 not rs3570595016. The authors need to correct this.
Response: We thank the Reviewer for his/her comments. We have corrected accordingly the text on page 14.
Reviewer #4:
Comment: Kirillov et al (The American Journal of Pathology, Vol. 185, 2015) mention that membrane-associated MMP-14 and its soluble zymogen substrate, MMP-2, have been found strongly up-regulated in IPF lung tissue. Besides, increased levels of active MMP-2 were reported in bronchoalveolar lavage fluids from IPF patients. These authors detected a markedly increased expression of MMP-14 protein in IPF lungs compared with normal and non-IPF lung tissue. Neither MMP-2 nor MMP-14 are discussed in this review.
Likewise, expression levels and localization of MMP and TIMP in IPF lungs have been reported to undergo substantial changes, with the levels of MMP1, MMP2, MMP9 and four TIMPs having also been found up-regulated (Mei et al, Frontiers in Pharmacology, 19 Jan 2022).
Response: We appreciate Reviewer’s comments. We have modified the text accordingly adding the evidence regarding the other MMPs and also the citations that were suggested to us.
Regarding TIMPs, we hadn’t previously included them as their role remains unclear in IPF. Therefore, according to reviewer’s comments, we added a new paragraph regarding their role as biomarkers.
Precisely, we added on page 7, Lines: 187-188, the following text.
‘In general, MMPs have been proposed to act as positive regulators of fibrogenesis in several studies.10,39,40’
Moreover, on page 8, Lines: 210-227
‘Moreover, other MMPs, such as MMP-2, MMP-9, MMP-14, MMP-19 have been also investigated as diagnostic and prognostic biomarkers in IPF.10,39,40,44 Indeed, MMP-2, but also MMP-14 are found strongly up-regulated in IPF lung tissue compared to non IPF patients.44 Increased levels of active MMP-2 have been also reported in BALF from IPF patients.44 Similarly, MMP-9 has been also found highly up-regulated in IPF lungs and specifically in myofibroblast foci enhancing fibroblast migration.45,46
Tissue inhibitors of metalloproteinases (TIMPs)
Tissue inhibitors of metalloproteinases control the catalytic action of MMPs and consist of four members (TIMP1-TIMP4).47,48 Their exact role in the pathogenesis of IPF but also as potential diagnostic or prognostic biomarkers remains still unclear.48 TIMPs levels have been found up-regulated in IPF lungs compared to controls. Particularly, TIMP-2 seem to be overexpressed in myofibroblast foci, while TIMP-3 is found in the thickened alveolar space.48 It has been proposed that TGF-b pathway interferes with TIMP-1 signaling.45,48 However, further studies are necessary to unravel the exact role of TIMPs in lung fibrosis.’

Reviewer 4 Report
Kirillov et al (The American Journal of Pathology, Vol. 185, 2015) mention that membrane-associated MMP-14 and its soluble zymogen substrate, MMP-2, have been found strongly up-regulated in IPF lung tissue. Besides, increased levels of active MMP-2 were reported in bronchoalveolar lavage fluids from IPF patients. These authors detected a markedly increased expression of MMP-14 protein in IPF lungs compared with normal and non-IPF lung tissue. Neither MMP-2 nor MMP-14 are discussed in this review.
Likewise, expression levels and localization of MMP and TIMP in IPF lungs have been reported to undergo substantial changes, with the levels of MMP1, MMP2, MMP9 and four TIMPs having also been found up-regulated (Mei et al, Frontiers in Pharmacology, 19 Jan 2022).
Author Response

(The authors gave the same response as above.)
